# AN ATTENTION-BASED DEEP NET FOR LEARNING TO RANK

## ABSTRACT

In information retrieval, learning to rank constructs a machine-based ranking model which given a query, sorts the search results by their degree of relevance or importance to the query. Neural networks have been successfully applied to this problem, and in this paper, we propose an attention-based deep neural network which better incorporates different embeddings of the queries and search results with an attention-based mechanism. This model also applies a decoder mechanism to learn the ranks of the search results in a listwise fashion. The embeddings are trained with convolutional neural networks or the word2vec model. We demonstrate the performance of this model with image retrieval and text querying data sets.

## 1 INTRODUCTION

Learning to rank applies supervised or semi-supervised machine learning to construct ranking models for information retrieval problems. In learning to rank, a query is given and a number of search results are to be ranked by their relevant importance given the query. Many problems in information retrieval can be formulated or partially solved by learning to rank. In learning to rank, there are typically three approaches: the pointwise, pairwise, and listwise approaches Liu (2011). The pointwise approach assigns an importance score to each pair of query and search result. The pairwise approach discerns which search result is more relevant for a certain query and a pair of search results. The listwise approach outputs the ranks for all search results given a specific query, therefore being the most general.

For learning to rank, neural networks are known to enjoy a success. Generally in such models, neural networks are applied to model the ranking probabilities with the features of queries and search results as the input. For instance, RankNet Burges et al. (2005) applies a neural network to calculate a probability for any search result being more relevant compared to another. Each pair of query and search result is combined into a feature vector, which is the input of the neural network, and a ranking priority score is the output. Another approach learns the matching mechanism between the query and the search result, which is particularly suitable for image retrieval. Usually the mechanism is represented by a similarity matrix which outputs a bilinear form as the ranking priority score; for instance, such a structure is applied in Severyn & Moschitti (2015).

We postulate that it could be beneficial to apply multiple embeddings of the queries and search results to a learning to rank model. It has already been observed that for training images, applying a committee of convolutional neural nets improves digit and character recognition Ciresan et al. (2011); Meier et al. (2011). From such an approach, the randomness of the architecture of a single neural network can be effectively reduced. For training text data, combining different techniques such as tf-idf, latent Dirichlet allocation (LDA) Blei et al. (2003), or word2vec Mikolov et al. (2013), has also been explored by Das et al. (2015). This is due to the fact that it is relatively hard to judge different models *a priori*. However, we have seen no literature on designing a mechanism to incorporate different embeddings for ranking. We hypothesize that applying multiple embeddings to a ranking neural network can improve the accuracy not only in terms of "averaging out" the error, but it can also provide a more robust solution compared to applying a single embedding.

For learning to rank, we propose the application of the attention mechanism Bahdanau et al. (2015); Cho et al. (2015), which is demonstrated to be successful in focusing on different aspects of the input so that it can incorporate distinct features. It incorporates different embeddings with weights changing over time, derived from a recurrent neural network (RNN) structure. Thus, it can help us

better summarize information from the query and search results. We also apply a decoder mechanism to rank all the search results, which provides a flexible list-wise ranking approach that can be applied to both image retrieval and text querying. Our model has the following contributions: (1) it applies the attention mechanism to listwise learning to rank problems, which we think is novel in the learning to rank literature; (2) it takes different embeddings of queries and search results into account, incorporating them with the attention mechanism; (3) double attention mechanisms are applied to both queries and search results.

Section 2 reviews RankNet, similarity matching, and the attention mechanism in details. Section 3 constructs the attention-based deep net for ranking, and discusses how to calibrate the model. Section 4 demonstrates the performance of our model on image retrieval and text querying data sets. Section 5 discusses about potential future research and concludes the paper.

## 2 LITERATURE REVIEW

To begin with, for RankNet, each pair of query and search result is turned into a feature vector. For two feature vectors $x_0 \in \mathbb{R}^{d_0}$ and $x_0' \in \mathbb{R}^{d_0}$ sharing the same query, we define $x_0 \prec x_0'$ if the search result associated with $x_0$ is ranked before that with $x_0'$, and *vice versa*. For $x_0$,

$$\begin{cases} x_1 = f(W_0 x_0 + b_0) \in \mathbb{R}^{d_1}, \\ x_2 = f(W_1 x_1 + b_1) \in \mathbb{R}^{d_2} = \mathbb{R}, \end{cases}$$

and similarly for $x_0'$. Here $W_l$ is a $d_{l+1} \times d_l$ weight matrix, and $b_l \in \mathbb{R}^{d_{l+1}}$ is a bias for $l = 0, 1$. Function $f$ is an element-wise nonlinear activation function; for instance, it can be the sigmoid function $\sigma(u) = e^u/(1 + e^u)$. Then for RankNet, the ranking probability is defined as $P(x_0 \prec x_0') = e^{x_2 - x_2'}/(1 + e^{x_2 - x_2'})$. Therefore the ranking priority of two search results can be determined with a two-layer neural network structure, offering a pairwise approach. A deeper application of RankNet can be found in Song et al. (2014), where a five-layer RankNet is proposed, and each data example is weighed differently for each user in order to adapt to personalized search. A global model is first trained with the training data, and then a different regularized model is adapted for each user with a validation data set.

A number of models similar to RankNet has been proposed. For instance, LambdaRank Burges et al. (2006) speeds up RankNet by altering the cost function according to the change in NDCG caused by swapping search results. LambdaMART Burges (2010) applies the boosted tree algorithm to LambdaRank. Ranking SVM Joachims (2002) applies the support vector machine to pairs of search results. Additional models such as ListNet Cao et al. (2007) and FRank Tsai et al. (2007) can be found in the summary of Liu (2011).

However, we are different from the above models not only because we integrate different embeddings with the attention mechanism, but also because we learn the matching mechanism between a query and search results with a similarity matrix. There are a number of papers applying this structure. For instance, Severyn & Moschitti (2015) applied a text convolutional neural net together with such a structure for text querying. For image querying, Wan et al. (2014) applied deep convolutional neural nets together with the OASIS algorithm Chechik et al. (2009) for similarity learning. Still, our approach is different from them in that we apply the attention mechanism, and develop an approach allowing both image and text queries.

We explain the idea of similarity matching as follows. We take a triplet $(q, r, r')$ into account, where $q$ denotes an embedding, i.e. vectorized feature representation of a query, and $(r, r')$ denotes the embeddings of two search results. A similarity function is defined as $S_W(q, r) = q^T W r$, and apparently $r \prec r'$ if and only if $S_W(q, r) > S_W(q, r')$.

Note that we may create multiple deep convolutional nets so that we obtain multiple embeddings for the queries and search results. Therefore, it is a question how to incorporate them together. The attention mechanism weighs the embeddings with different sets of weights for each state $t$, which are derived with a recurrent neural network (RNN) from $t = 1$ to $t = T$. Therefore, for each state $t$, the different embeddings can be "attended" differently by the attention mechanism, thus making the model more flexible. This model has been successfully applied to various problems. For instance, Bahdanau et al. (2015) applied it to neural machine translation with a bidirectional recurrent neural network. Cho et al. (2015) further applied it to image caption and video description generation with

convolutional neural nets. Vinyals et al. (2015) applied it for solving combinatorial problems with the sequence-to-sequence paradigm.

Note that in our scenario, the ranking process, i.e. sorting the search results from the most related one to the least related one for a query, can be modeled by different "states." Thus, the attention mechanism helps incorporating different embeddings along with the ranking process, therefore providing a listwise approach. Below we explain our model in more details.

# 3 MODEL AND ALGORITHM

## 3.1 INTRODUCTION TO THE MODEL

Both queries and search results can be embedded with neural networks. Given an input vector $x_0$ representing a query or a search result, we denote the $l$-th layer in a neural net as $x_l \in \mathbb{R}^{d_l}$, $l = 0, 1, \ldots, L$. We have $x_{l+1} = f(W_l x_l + b_l)$, $l = 0, 1, \ldots, L-1$ where $W_l$ is a $d_{l+1} \times d_l$ weight matrix, $b_l \in \mathbb{R}^{d_{l+1}}$ is the bias, and $f$ is a nonlinear activation function. If the goal is classification with $C$ categories, then $(P(y = 1), \ldots, P(y = C)) = softmax(W_L x_L + b_L)$, where $y$ is a class indicator, and $softmax(u) = (e^{u_1}/\sum_{i=1}^{d} e^{u_i}, \ldots, e^{u_d}/\sum_{i=1}^{d} e^{u_i})$ for $u \in \mathbb{R}^d$.

From training this model, we may take the softmax probabilities as the embedding, and create different embeddings with different neural network structures. For images, convolutional neural nets (CNNs) LeCun et al. (1998) are more suitable, in which each node only takes information from neighborhoods of the previous layer. Pooling over each neighborhood is also performed for each layer of a convolutional neural net.

With different networks, we can obtain different embeddings $c^1, \ldots, c^M$. In the attention mechanism below, we generate the weights $\alpha_t$ with an RNN structure, and summarize $c_t$ in a decoder series $z_t$,

$$\begin{cases} e_{tm} = f_{ATT}(z_{t-1}, c^m, \alpha_{t-1}), \ m = 1, \ldots, M, \\ \alpha_t = softmax(e_t), \\ c_t = \sum_{m=1}^{M} \alpha_{tm} c^m, \\ z_t = \phi_\theta(z_{t-1}, c_t). \end{cases}$$

Here $f_{ATT}$ and $\phi_\theta$ are chosen as $\tanh$ layers in our experiments. Note that the attention weight $\alpha_t$ at state $t$ depends on the previous attention weight $\alpha_{t-1}$, the embeddings, and the previous decoder state $z_{t-1}$, and the decoder series $z_t$ sums up information of $c_t$ up to state $t$.

As aforementioned, given multiple embeddings, the ranking process can be viewed as applying different attention weights to the embeddings and generating the decoder series $z_t$, offering a listwise approach. However, since there are features for both queries and search results, we consider them as separately, and apply double attention mechanisms to each of them. Our full model is described below.

## 3.2 MODEL CONSTRUCTION

For each query $Q$, we represent $M$ different embeddings of the query as $q = (q_1, \ldots, q_M)$, where each $q_m$ is multi-dimensional. For the search results $R_1, \ldots, R_T$ associated with $Q$, which are to be ranked, we can map each result into $N$ different embeddings with a same structure, and obtain $r_t = (r_{t1}, \ldots, r_{tN})$ as the embeddings of $R_t$, $t = 1, \ldots, T$. Each $r_{tn}$ is also multi-dimensional. All these embeddings can either correspond to raw data or they can be the output of a parametric model, i.e. the output of a CNN in the case of images. In the latter case, they are trained jointly with the rest of the model.

Below we represent queries or search results in their embeddings. We assume that given the query $q$, the results $r_1, \ldots r_T$ are retrieved in the order $\tilde{r}_1, \ldots, \tilde{r}_T$ in equation (12) below. Note that $\{r_1, \ldots, r_T\} = \{\tilde{r}_1, \ldots, \tilde{r}_T\}$.

We observe that both the query $q$ and the results $r_1, \ldots, r_T$ can be "attended" with different weights. To implement the attention-based mechanism, we assign different weights for different components of for each different $t = 1, \ldots, T$. Specifically, we assign the attention weights $\alpha_1, \ldots, \alpha_T$ for the

query, and $\beta_1, \ldots, \beta_T$ for the search results. To achieve this, we need to assign the pre-softmax weights $e_t$ and $f_t$, $t = 1, \ldots, T$. Therefore we first let

$$\begin{cases} e_{tm} = e_{ATT}(z_{t-1}, q_m, g(r_1, \ldots, r_T), \alpha_{t-1}, \beta_{t-1}), \\ f_{tn} = f_{ATT}(z_{t-1}, q, h_n(r_1, \ldots, r_T), \alpha_{t-1}, \beta_{t-1}), \\ m = 1, \ldots, M, \ n = 1, \ldots, N, \ t = 1, \ldots, T. \end{cases} \quad (1)$$

Note that there are two different attention functions. While they both consider the previous weights $\alpha_{t-1}$ and $\beta_{t-1}$, the pre-softmax weight $e_{tm}$ considers the $m$-th component of the query $q_m$ and all information from the search results, and $f_{tn}$ considers the opposite. In our experiments, $g$ takes an average while $h$ averages over the $n$-th embedding of each result. Here $z_{t-1}$ is the value of the decoder at state $t-1$. Letting $e_t = (e_{t1}, \ldots, e_{tM})$, $f_t = (f_{t1}, \ldots, f_{tN})$, we impose the softmax functions for $t = 1, \ldots, T$,

$$\alpha_t = softmax(e_t), \ \beta_t = softmax(f_t). \quad (2)$$

The attention weights $\alpha_t$ and $\beta_t$ can then be assigned to create the context vectors $c_t$ for the query and $\bar{d}_t$ for the search results, which are defined as follows,

$$c_t = \sum_{m=1}^{M} \alpha_{tm} q_m, \ \bar{d}_t = \sum_{n=1}^{N} \beta_{tn} h_n(r_1, \ldots, r_T). \quad (3)$$

With the context vectors $c_t$ and $\bar{d}_t$ depending on the state $t$, we can finally output the decoder $z_t$,

$$z_t = \phi_\theta(z_{t-1}, c_t, \bar{d}_t), \ t = 1, \ldots, T. \quad (4)$$

Equations (8-11) are carried out iteratively from $t = 1$ to $t = T$. The hidden states $z_1, \ldots, z_T$, together with the context vectors $c_1, \ldots, c_T$, are used to rank results $r_1, \ldots, r_T$. For each $r_t$, we define a context vector $d_{t,t'}$ which can be directly compared against the context vector $c_t$ for queries,

$$d_{t,t'} = \sum_{n=1}^{N} \beta_{tn} r_{t'n}, \ t = 1, \ldots, T. \quad (5)$$

This context vector $d_{t,t'}$, unlike $\bar{d}_t$, is not only specific for each state $t$, but also specific for each result $r_{t'}$, $t' = 1, \ldots, T$. Now suppose in terms of ranking that $\tilde{r}_1, \ldots, \tilde{r}_{t-1}$ have already been selected. For choosing $\tilde{r}_t$, we apply the softmax function for the similarity scores $s_{t,t'}$ between the query $q$ and each result $r_{t'}$.

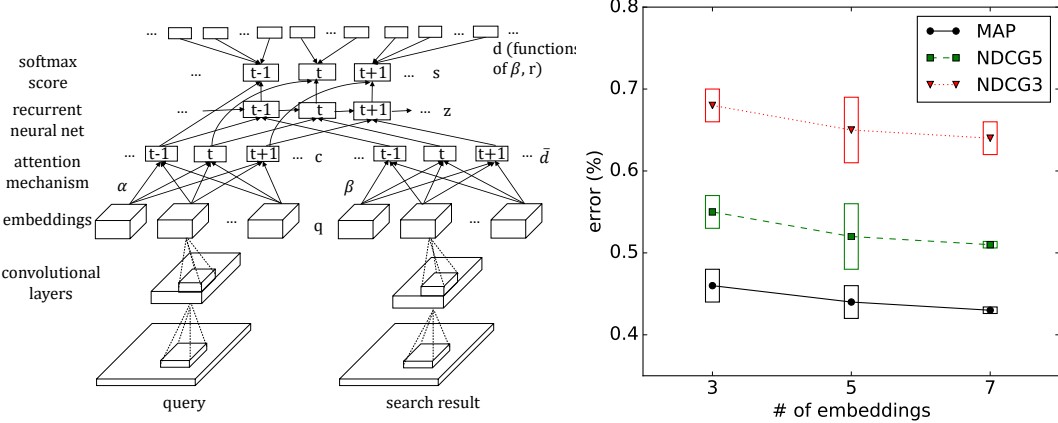

softmax score

recurrent neural net

attention mechanism

embeddings

convolutional layers

query          search result

Figure 1: The attention-based model for image retrieval.

Figure 2: The error rates of AttRN-HL given different numbers of embeddings with 95% upper and lower confidence bounds.

Thus we have

$$\begin{cases} P(\tilde{r}_t = r_{t'} | \tilde{r}_1, \ldots, \tilde{r}_{t-1}) \propto softmax(s_{t,r_{t'}}), \\ s_{t,r_{t'}} = d_{t,t'}^T W c_t + d_{t,t'}^T V z_t, \\ r_{t'} \in \{r_1, \ldots, r_T\} \backslash \{\tilde{r}_1, \ldots, \tilde{r}_{t-1}\}. \end{cases} \quad (6)$$

Therefore $\tilde{r}_1, \ldots, \tilde{r}_T$ can be retrieved in a listwise fashion. Equations (1-6) complete our model, which is shown in Figure 1 in the case of image retrieval.

### 3.3 Model Calibration

To calibrate the parameters of this model, we apply the stochastic gradient descent algorithm which trains a minibatch, a subsample of the training data at each iteration. To rank search results for the testing data, we apply the beam search algorithm Reddy (1977) with a beam width of 3, which keeps a number of paths, i.e. sequences of $\tilde{r}_1, \ldots, \tilde{r}_t$ at state $t$, of the highest log-likelihood. Other paths are trimmed at each state $t$, and finally the path with the highest log-likelihood is chosen.

Alternatively, we may consider the hinge loss function to replace the softmax function. Comparing a potentially chosen result $\tilde{r}_t$ against any other unranked result $r_{t'}$, with $r_{t'} \in \{r_1, \ldots, r_T\} \backslash \{\tilde{r}_1, \ldots, \tilde{r}_{t-1}\}$, we apply the hinge loss function, similar to Chechik et al. (2009) $\mathcal{L}(\tilde{r}_t, r_{t'}) = \max\{0, 1 - s_{t,\tilde{r}_t} + s_{t,r_{t'}}\}$. When training the model, for a single query $q$ and all related search results, we apply the stochastic gradient descent algorithm to minimize $\sum_{t=1}^{T} \sum_{r_{t'}} \mathcal{L}(\tilde{r}_t, r_{t'})$. For this method, we maximize $s_{t,r_{t'}}$ at each step, i.e. apply beam search with a beam width of 1.

## 4 Data Studies

We perform studies on three datasets: MNIST presented in Section 4.1, CIFAR-10 listed in Appendix, and 20 Newsgroups in Section 4.2.

### 4.1 The MNIST Data Set

The MNIST data set contains handwritings of 0-9 digits. There are 50,000 training, 10,000 validation, and 10,000 testing images. We take each image among them as a query. For each query image, we construct the set of retrieved images as follows. We first uniformly at random sample a random integer $k$ from $\{1, 2, \ldots, 9\}$. Next we select $k$ related random images (an image is related if it is of the same digit as the query image). Finally, we randomly select 30-$k$ unrelated images (an image is unrelated if it corresponds to a different digit than the query image). The order is imposed so that images of the same digit are considered as related and images of different digits not related.

For this data set, we pretrain 5 convolutional neural networks based on different $L^2$ regularization rates for all layers and Dropout Srivastava et al. (2014) regularization for fully connected (FC) layers with respect to the values shown in Table 1. We vary the regularization rate because it can be expensive to determine a most suitable regularization rate in practice, and therefore applying the attention mechanism to different regularization rates can reduce the impact of improper regularization rates and find a better model.

Table 1: Different regularization rates applied to MNIST.

| | | | | | |
|---|---|---|---|---|---|
| Dropout's $p$ for Conv. | 1 | 1 | 1 | 1 | 1 |
| Dropout's $p$ for FC | 1 | 1 | 0.9 | 1 | 0.9 |
| $L^2$'s $\lambda$ for All | 0 | 1e-5 | 1e-5 | 1e-4 | 1e-4 |

Table 2: Errors of "mean" and "max" pooling as for MNIST.

| error | MAP | NDCG$_3$ | NDCG$_5$ |
|---|---|---|---|
| AttRN-HL-mean | 0.44% | 0.65% | 0.52% |
| sd. | (0.01%) | (0.02%) | (0.02%) |
| AttRN-HL-max | 0.44% | 0.66% | 0.53% |
| sd. | (0.01%) | (0.02%) | (0.02%) |

The pretrained models are standard digit classification. The softmax layers of the five models as embeddings are plugged into our full model (where the CNNs are trained together with the rest). We initialize the similarity matrix $W$ with the identity matrix and the remaining parameters with zeros. Our model is trained given such embeddings with a batch size of 100, a learning rate of 0.001, and 20 epochs. We choose the best epoch using the validation data set.

We compare our algorithm against OASIS Chechik et al. (2009), Ranking SVM, and LambdaMART, in the setting of image retrieval problems Wan et al. (2014); Wang et al. (2014); Wu et al. (2013). For OASIS, we set the batch size and maximum learning rate to be the same as our model. For Ranking SVM and LambdaMART, we combine the embeddings of a query and a search result to

create features, apply sklearn and pyltr in Python, and use $NDCG_5$ and 100 trees for LambdaMART. To compare the results, we apply MAP and $NDCG_p$ Liu (2011) as the criteria, and calculate the standard deviations averaged over five runs for each model. The error rates, which are *one minus* MAPs and NDCGs, are shown in Table 3. The lowest error rates are in bold. The top row shows the average value while the bottom one (sd.) exhibits the standard deviation for 5 randomized runs.

We name our model as an "Attention-based Ranking Network" (AttRN). In Table 3, "AttRN-SM" or "AttRN-HL" denote our model with softmax or hinge loss ranking functions. "OASIS-1" or "OASIS-5" denote, respectively, OASIS trained with the first set of embeddings, or the average of all five sets of embeddings. For Ranking SVM and LambdaMART, we use all embeddings. "RSVM-100k" means that $10^5$ pairs are sampled for the SVM, "$\lambda$MART-2%" means that 2% of all feature vectors are sampled for each tree in LambdaRank, and so forth.

Table 3: Errors of different image retrieval methods for MNIST.

| error | OASIS-1 | OASIS-5 | AttRN-SM | AttRN-HL |
|---|---|---|---|---|
| MAP | 0.55% | 0.47% | 0.46% | **0.44%** |
| sd. | (0.02%) | (0.01%) | (0.01%) | (0.01%) |
| $NDCG_3$ | 0.81% | 0.70% | 0.68% | **0.65%** |
| sd. | (0.02%) | (0.01%) | (0.01%) | (0.02%) |
| $NDCG_5$ | 0.65% | 0.57% | 0.55% | **0.52%** |
| sd. | (0.02%) | (0.01%) | (0.02%) | (0.02%) |
| error | RSVM-100k | RSVM-500k | $\lambda$MART-2% | $\lambda$MART-5% |
| MAP | 0.52% | 0.54% | 0.87% | 0.88% |
| sd. | (0.02%) | (0.01%) | (0.02%) | (0.03%) |
| $NDCG_3$ | 0.74% | 0.75% | 0.93% | 0.96% |
| sd. | (0.03%) | (0.01%) | (0.01%) | (0.02%) |
| $NDCG_5$ | 0.61% | 0.62% | 0.89% | 0.91% |
| sd. | (0.02%) | (0.01%) | (0.02%) | (0.03%) |

From Table 3, we observe that AttRN-HL outperforms other methods, and AttRN-SM achieves the second best. Moreover, OASIS-5 clearly outperforms OASIS-1, which demonstrates the benefit of applying multiple embeddings. Ranking SVM and LambdaMART do not seem very suitable for the data, and increasing the sample size does not seem to improve their performance.

It needs to be investigated how sensitive AttRN-HL is against the number of embeddings. In Figure 2, we show the error rates of AttRN-HL with different numbers of embeddings. Here we apply regularization parameters (1, 0.8, 1e-4) and (1, 0.8, 2e-4) to the new embeddings. From Figure 2, the error rates steadily decrease as the number of embeddings increases, which is consistent with the intuition that adding more information to the model should yield better results. We also observe that the error rates tend to stabilize as the number of embeddings increases.

However, while applying 7 embeddings slightly lowers the error rates as shown in Figure 2, the training time is roughly 33% longer than for 5 embeddings. Therefore we consider applying 7 embeddings to be unnecessary, and consider the current practice to be representative of AttRN.

Table 2 shows the differences in error rates by changing the pooling functions $g$ and $h_n$ in (7) from "mean" to "max." We observe that the changes are very small, which means that AttRN is quite robust to such changes.

Table 4: Errors of "mean" and "max" pooling as for MNIST.

| error | MAP | $NDCG_3$ | $NDCG_5$ |
|---|---|---|---|
| AttRN-HL-mean | 0.44% | 0.65% | 0.52% |
| sd. | (0.01%) | (0.02%) | (0.02%) |
| AttRN-HL-max | 0.44% | 0.66% | 0.53% |
| sd. | (0.01%) | (0.02%) | (0.02%) |

Table 4 shows the changes in weights of the convolutional neural networks. The unregularized CNN is used as an example. The $L^2$ norms sum over the weights of each layer and are averaged over 5 randomized runs. They tend to increase after AttRN is trained, possibly because it requires the weights to be larger to adjust for the ranking mechanism.

Table 5: $L^2$ norms of each layer in the convolutional neural net before and after training AttRN as for MNIST.

| $L^2$ norm | 1st Conv. | 2nd Conv. | FC | Softmax |
|---|---|---|---|---|
| Before | 4.99 | 9.66 | 25.74 | 7.15 |
| sd. | (0.05) | (0.03) | (0.02) | (0.04) |
| After | 5.26 | 9.91 | 25.90 | 7.81 |
| sd. | (0.05) | (0.03) | (0.02) | (0.04) |

query/images considered related / unrelated

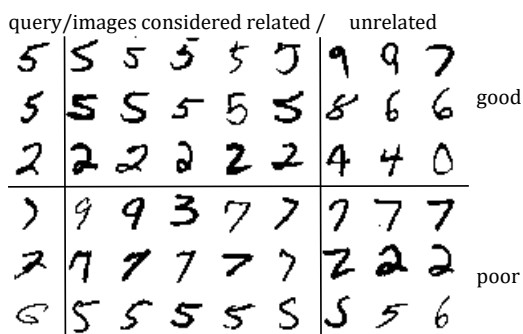

good

poor

Figure 3: Actual query images and the 1st to 8th retrieved images in the MNIST data set.

query/ considered related / unrelated

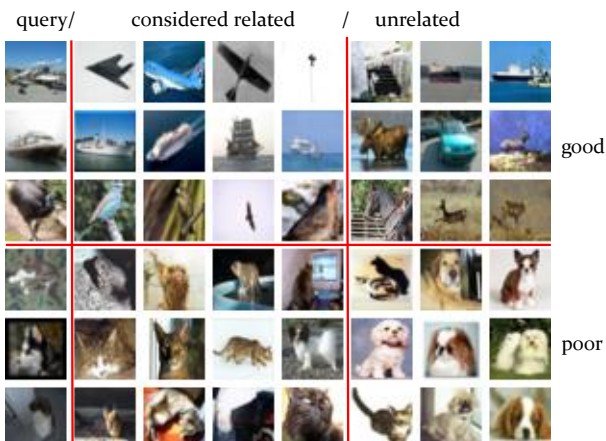

good

poor

Figure 4: Actual query images and the 1st to 7th retrieved images in the CIFAR-10 data set.

Next we display three good cases of retrieval together with three poorer ones from AttRN-HL in terms of the actual images, which are shown in Figure 3. We observe that the poorer cases of retrieval in the MNIST data set are basically due to the distortion of the query images.

### 4.2 THE 20 NEWSGROUPS DATA SET

The 20 Newsgroups data set contains online news documents with 20 different topics. There are 11,293 training documents, from which 1,293 are held out for validation, and 7,528 testing documents. The topics can be categorized into 7 superclasses: religion, computer, "for sale," cars, sports, science, and politics. We consider each document as a query, and randomly choose 30 retrieved documents as follows: the number of documents of the same topic is uniformly distributed from 3 to 7, the number of documents of the same superclass but different topics is also uniformly distributed from 3 to 7, and the remaining documents are of different superclasses. We impose the order on different documents so that: (1) documents with the same topic are considered to be related, and otherwise not; (2) documents with the same superclass are considered to be related, and otherwise not. We train our model with the first type of order, and calculate error rates for both orders.

We pretrain the documents with three different word2vec-type Mikolov et al. (2013) models: CBOW, skip-gram, and GLOVE Pennington et al. (2014). We note that all these techniques apply neighboring words to predict word occurrences with a neural network based on different assumptions. Therefore, applying the attention mechanism to incorporate the three models can weaken the underlying distributional assumptions and result in a more flexible word embedding structure.

We apply text8 as the training corpus for CBOW and skip-gram, and download the pretrained weights for GLOVE from Wikipedia 2014 and Gigaword 5. Each vector representing a word of dimension 300. For each document, we represent it as a tf-idf vector Aggarwal & Zhai (2012) and apply the word2vec weights to transform it into a 300-dimensional vector, which is the same as VecAvg in Socher et al. (2013). For this instance, the embeddings are fixed and not modified during the training

process of our model. We also impose a softmax layer corresponding to the 20 classes on top of word2vec and use the resulting final weights as pretrained weights to AttRN. We compare our model against OASIS as well as Ranking SVM and LambdaMART. The results of RankNet are not shown here because we have found it to be unsuitable for this data set.

For AttRN and OASIS, we apply a batch size of 100 and a learning rate of 0.0001, and 50 epochs. Other computational details are the same as in the previous data sets. Table 8 considers any two documents of the same topic to be related, while Table 9 considers any two documents of the same superclass to be related.

Table 6: Errors of different text retrieval methods for 20 Newsgroups with regard to different topics.

| error | OASIS-1 | OASIS-3 | AttRN-SM | AttRN-HL |
|---|---|---|---|---|
| MAP | 15.56% | 15.07% | **14.78%** | 14.79% |
| sd. | (0.01%) | (0.01%) | (0.01%) | (0.004%) |
| $NDCG_3$ | 17.83% | 17.25% | **16.87%** | 16.93% |
| sd. | (0.02%) | (0.02%) | (0.02%) | (0.01%) |
| $NDCG_5$ | 17.16% | 16.59% | **16.28%** | 16.29% |
| sd. | (0.02%) | (0.02%) | (0.01%) | (0.02%) |
| error | RSVM-20k | RSVM-100k | λMART-2% | λMART-5% |
| MAP | 23.24% | 19.98% | 32.90% | 31.58% |
| sd. | (0.60%) | (0.10%) | (1.20%) | (0.87%) |
| $NDCG_3$ | 24.35% | 21.20% | 34.62% | 33.55% |
| sd. | (0.20%) | (0.05%) | (1.26%) | (0.88%) |
| $NDCG_5$ | 24.73% | 21.37% | 35.33% | 34.04% |
| sd. | (0.48%) | (0.08%) | (1.27%) | (0.95%) |

Table 7: Errors of different text retrieval methods for 20 Newsgroups with regard to different superclasses.

| error | OASIS-1 | OASIS-3 | AttRN-SM | AttRN-HL |
|---|---|---|---|---|
| MAP | 30.58% | 30.15% | **30.14%** | 30.19% |
| sd. | (0.02%) | (0.01%) | (0.01%) | (0.04%) |
| $NDCG_3$ | 13.59% | 13.14% | **12.93%** | 12.95% |
| sd. | (0.02%) | (0.02%) | (0.02%) | (0.01%) |
| $NDCG_5$ | 18.36% | 17.92% | 17.82% | **17.79%** |
| sd. | (0.03%) | (0.01%) | (0.01%) | (0.01%) |
| error | RSVM-20k | RSVM-100k | λMART-2% | λMART-5% |
| MAP | 35.87% | 34.03% | 41.52% | 40.27% |
| sd. | (0.30%) | (0.06%) | (0.91%) | (0.55%) |
| $NDCG_3$ | 19.61% | 16.83% | 29.35% | 28.34% |
| sd. | (0.25%) | (0.04%) | (1.24%) | (0.63%) |
| $NDCG_5$ | 24.92% | 22.10% | 34.05% | 32.65% |
| sd. | (0.38%) | (0.08%) | (1.14%) | (0.70%) |

Tables 8 and 9 show that AttRN-SM is the most suitable for 20 Newsgroups while AttRN-HL achieves the second best, and is relatively close. We also observe that MAP for superclasses is too strict for this data set, because all documents of the same superclass are taken into account. NDCG scores are more realistic because a threshold is given and less related search results are down-weighted.

To conclude, AttRN-HL has the most robust performance across all three data sets, and is thus the recommended choice. Moreover, for the problems considered in this work, OASIS is more suitable than Ranking SVM and LambdaMART, and thus a competitive baseline.

## 5 CONCLUSION

In this paper, we proposed a new neural network for learning-to-rank problems which applies the attention mechanism to incorporate different embeddings of queries and search results, and ranks the search results with a listwise approach. Data experiments show that our model yields improvements over state-of-the-art techniques. For the future, it would be of interest to consider improving the RNN structure in the attention mechanism, and tailoring the embedding part of the neural network to this problem.

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

# A  APPENDIX

## A.1  THE CIFAR-10 DATA SET

The CIFAR-10 data set contains images of 10 different classes. There are 50,000 training images, from which we take 5,000 images for validation, and 10,000 testing images. Again we took each image as a query, and randomly created 30 search results with the number of related images uniformly distributed from 1 to 9. The order is imposed in the same way as MNIST except that it is based on different classes. We use 5 convolutional neural networks in the same way as MNIST based on regularization rates shown in Table 5.

Table 8: Different regularization rates applied to CIFAR-10.

| Dropout's $p$ for Conv. | 0.25 | 0.25 | 0.25 | 0.1 | 0.1 |
|---|---|---|---|---|---|
| Dropout's $p$ for FC | 0.5 | 0.3 | 0.1 | 0.3 | 0.1 |
| $L^2$'s $\lambda$ | 5e-4 | 5e-4 | 5e-4 | 5e-4 | 5e-4 |

We apply a batch size of 50, a learning rate of 0.0005 and 20 epochs to train AttRN. Other computational details are the same as in the MNIST data set. The error rates and the standard deviations are shown in Table 6. AttRN-HL again achieves the best performance followed by AttRN-SM and OASIS-5 in terms of MAP.

Table 9: Errors of different image retrieval methods for CIFAR-10.

| error | OASIS-1 | OASIS-5 | AttRN-SM | AttRN-HL |
|---|---|---|---|---|
| MAP | 16.46% | 13.72% | 13.65% | **13.41%** |
| sd. | (0.08%) | (0.08%) | (0.06%) | (0.07%) |
| NDCG$_3$ | 19.19% | 15.95% | 15.93% | **15.61%** |
| sd. | (0.11%) | (0.08%) | (0.06%) | (0.08%) |
| NDCG$_5$ | 17.94% | 14.86% | 14.80% | **14.55%** |
| sd. | (0.06%) | (0.08%) | (0.08%) | (0.07%) |
| error | RSVM-100k | RSVM-500k | λMART-2% | λMART-5% |
| MAP | 13.74% | 13.77% | 18.49% | 19.27% |
| sd. | (0.20%) | (0.13%) | (0.06%) | (0.14%) |
| NDCG$_3$ | 15.91% | 15.87% | 19.34% | 20.05% |
| sd. | (0.20%) | (0.15%) | (0.22%) | (0.46%) |
| NDCG$_5$ | 14.78% | 14.81% | 18.71% | 19.28% |
| sd. | (0.20%) | (0.14%) | (0.11%) | (0.26%) |

Table 7 also shows that replacing the pooling functions from "mean" to "max" again yields very small changes in error rates, which means that AttRN is quite robust.

Table 10: Errors of "mean" and "max" pooling as for CIFAR-10.

| error | MAP | NDCG$_3$ | NDCG$_5$ |
|---|---|---|---|
| AttRN-HL-mean | 13.41% | 15.61% | 14.55% |
| sd. | (0.07%) | (0.08%) | (0.07%) |
| AttRN-HL-max | 13.40% | 15.61% | 14.54% |
| sd. | (0.07%) | (0.08%) | (0.07%) |

Again we display three good cases of retrieval and three poorer ones from AttRN-HL in terms of the actual images, shown in Figure 4 below. We observe that the images considered as related tend to be in the same superclass as the query, but may be in a different class due to the variation within each class of images and blurred images.

