# OpenReview forum: "AN ATTENTION-BASED DEEP NET FOR LEARNING TO RANK"
_ICLR.cc/2020/Conference — Reject_

### Official Review · AnonReviewer1 · 2019-10-22
**Official Blind Review #1**

**Rating:** 1

**Review:**

In this paper, the authors propose to use attention to combine multiple input representations for both query and search results in the learning to rank task. When these representations are embeddings from differentiable functions, they can be jointly learned with the neural network which predicts rankings. A limited set of experiments suggest the proposed approach very mildly outperforms benchmark approaches.

Major comments

To the best of my knowledge, this is the first paper to apply attention to the learning to rank problem. However, the main methodological innovation seems to be the use of attention to create and train an ensemble of models; this has been previously explored in the literature (e.g., [Kim et al., ECCV 2018]).

The paper is also missing important context in that it omits developments in using deep learning for the learning to rank problem (e.g., [Pang et al., CIKM 2017; Ai et al., WWW 2018]). The experimental evaluation does not include any other deep methods; thus, it is not clear if the (very minor) improvement in performance are due to the deep models or the proposed attention approach.

The datasets used in the experiments are not appropriate for evaluating learning to rank algorithms. A variety of learning to rank datasets are available, and these should be used rather than (or in addition to) the toy datasets considered here. Examples: http://arogozhnikov.github.io/2015/06/26/learning-to-rank-software-datasets.html, http://quickrank.isti.cnr.it/istella-dataset/, https://www.cl.uni-heidelberg.de/statnlpgroup/nfcorpus/,

Minor comments

Concerning Section 3.3, in what sense is SGD used to “calibrate” the model? It seems as though the authors just mean it is used to “train” the model. However, is there some other meaning of calibration (e.g., in the sense of a Brier score) here?

In Table 1, what is the meaning of a dropout p value of 1? In most deep learning frameworks (e.g., Keras and PyTorch), this would mean all nodes are dropped out.

In what sense are the “5 randomized runs” for the experiments randomized? Are different train, test splits used? or just different random seeds? or something else?

How is it that the error rates are higher when using superclasses for evaluation?

Typos, etc.

The paper has several significant problems with the “\cite”s and “\ref”s in the paper. First, the “\cite”s should presumably be “\citep”s or something since the references are not set off from the rest of the text. Second, the paper includes references to equation numbers which are not present in the paper, such as “equation (12)”. It seems as the equations are in the paper, but are included in some unnumbered environment (“\begin{align*}” or some such). This makes it very difficult to track down to which equations the authors intend to refer. Third, the reference numbers to figures and tables in the text is wrong. For example, the text refers to “Tables 8 and 9” for 20 newsgroups (at the end of Section 4). Clearly, this is supposed to be Tables 6 and 7. It seems like the authors moved the CIFAR-10 discussion to the appendix but did not update the references in the text.

Tables 2 and 4 are exactly the same.

Figure 4 is not referenced in the text.

It would be helpful to put Figure 2 a bit closer to where it is discussed in the text.

The references are not consistently formatted.

“components of for each” -> “components for each”

Please define acronyms like MAP at least once.


**Experience Assessment:**

I have read many papers in this area.

**Review Assessment: Checking Correctness Of Derivations And Theory:**

I assessed the sensibility of the derivations and theory.

**Review Assessment: Checking Correctness Of Experiments:**

I carefully checked the experiments.

**Review Assessment: Thoroughness In Paper Reading:**

I read the paper at least twice and used my best judgement in assessing the paper.

---

### Official Review · AnonReviewer3 · 2019-10-24
**Official Blind Review #3**

**Rating:** 1

**Review:**

This paper proposes to use attention mechanism for combining different embeddings of the queries and search results. Besides, a decoder mechanism is used to do listwise ranking for the results. The experiments show that the proposed approach outperforms some classic learning-to-rank baselines.

This paper is below the bar of acceptance for the following reasons:

1.	Limited technical contribution: some previous papers have explored the idea of learning attention weights for combining different embeddings, and simply applying this idea to learning-to-rank application does not seem to be a big contribution.

2.	Choice of datasets: the datasets used in this paper are typically used for tesing classification models rather than ranking models. In these datasets, for each query image/doc, there are many images/docs of the same class that could be considered relevant, which makes the ranking task less challenging. Since the paper focuses on learning-to-rank problem, probably the authors should consider include more datasets dedicated to learning-to-rank problems.

3.	Insufficient baselines: the baseline methods used in the paper are not very recent (e.g., OASIS, RankSVM and LambdaMart have been proposed for more than 10 years). There have been many neural-network based retrieval/ranking methods proposed in the past 5 years. Hence, the experimental results could be more convincing if the paper include more

4.	Lack of justification for the model architecture: some design choices of the model are not well-motivated/justified. For example, how does the decoder mechanism using multiple states in the model (listwise) help improve the ranking results compared to pairwise ranking? Ablation study could help whether such decoder mechanism help show the usefulness of this module.

5.	Parameter sensitivity study: study on how hyper-parameter values affects the model performance could also help.


**Experience Assessment:**

I have published one or two papers in this area.

**Review Assessment: Checking Correctness Of Derivations And Theory:**

I assessed the sensibility of the derivations and theory.

**Review Assessment: Checking Correctness Of Experiments:**

I assessed the sensibility of the experiments.

**Review Assessment: Thoroughness In Paper Reading:**

I read the paper at least twice and used my best judgement in assessing the paper.

---

### Official Review · AnonReviewer2 · 2019-10-28
**Official Blind Review #2**

**Rating:** 1

**Review:**

The paper proposed an attention-based deep neural network for implementing 'learning to rank' algorithm. Particularly, the proposed method implements a listwise approach which outputs the ranks for all search results given a query. The search results are claimed to be sorted by their degree of relevance or importance to the query. However, it is not clear to me how the ranking was decided in equation 6 by the softmax function. For example, as per section 4, the documents of the same topic are considered related, then how the proposed model was trained with one document having higher relevance than others in the same topic category.

There are other confusions that need to be addressed for better understanding. For example, how softmax probabilities can be used as an embedding as described in the line: “From training this model, we may take the softmax probabilities as the embedding, and create different embeddings with different neural network structures. ” Also, what does the line means: “the number of documents of the same topic is uniformly distributed from 3 to 7, the number of documents of the same superclass but different topics is also uniformly distributed from 3 to 7, and the remaining documents are of different super classes.”


**Experience Assessment:**

I have published one or two papers in this area.

**Review Assessment: Checking Correctness Of Derivations And Theory:**

I assessed the sensibility of the derivations and theory.

**Review Assessment: Checking Correctness Of Experiments:**

I assessed the sensibility of the experiments.

**Review Assessment: Thoroughness In Paper Reading:**

I read the paper at least twice and used my best judgement in assessing the paper.

---

### Decision · Program_Chairs · 2019-12-19

**Decision:**

Reject

**Comment:**

All three reviewers felt the paper should be rejected and no rebuttal was offered. So the paper is rejected.